# Molecular Stratification of Renal Cancer Reveals Prognostic Biomarkers and Therapeutic Pathways

## Abstract

Kidney renal clear cell carcinoma (KIRC) is the most common subtype of renal cancer, yet the discovery of robust prognostic biomarkers has been hindered by its profound molecular heterogeneity, complex tumor microenvironment, and metabolic rewiring. Here, we present an integrative transcriptomic analysis of 314 KIRC patients to uncover molecular subtypes and biomarker signatures with clinical relevance. Using an iterative survival-guided feature selection approach, we refined 1,000 highly variable genes into a compact 76-gene signature that enabled unsupervised clustering into two prognostically distinct subgroups. Patients in the high-risk subgroup exhibited significantly poorer overall survival (log-rank p $= 4.5 \times 10^{-4}$) and elevated event rates compared to the low-risk group. Differential expression analysis revealed 2,927 subtype-specific genes, of which 70% demonstrated significant associations with survival in univariate Cox regression. Functional enrichment highlighted convergence on cancer-associated pathways, including TOR signaling, regulation of macroautophagy, and negative regulation of catabolic processes, implicating both canonical oncogenic drivers (e.g., PIK3CA, EIF4EBP2, PRKAA2) and modulators of cellular homeostasis (e.g., UBR1, MTM1). Together, these findings establish a refined prognostic biomarker framework for KIRC, define clinically relevant molecular subtypes, and reveal pathway-level vulnerabilities that may be exploited for therapeutic intervention.

## 1 Introduction

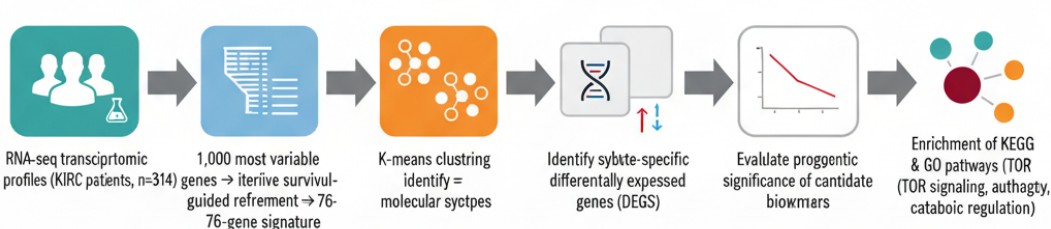

Figure 1: Study workflow pipeline.

Kidney renal clear cell carcinoma (KIRC), also known as clear cell renal cell carcinoma (ccRCC), represents the most prevalent subtype of renal cell carcinoma, accounting for 75–85% of kidney cancers and approximately 403,000 new cases worldwide annually [1, 2]. KIRC is characterized by distinctive genetic alterations, particularly VHL (Von Hippel–Lindau) inactivation, which drives angiogenesis and immune escape mechanisms, leading to the characteristic clear cytoplasmic appearance due to lipid and glycogen accumulation [2, 3]. However, the discovery of robust biomarkers for

Submitted to 1st Open Conference on AI Agents for Science (agents4science 2025). Do not distribute.

KIRC remains exceptionally challenging. The disease exhibits remarkable inter- and intratumoral heterogeneity, complicating the identification of universally applicable biomarkers [1, 4, 5]. The complex and immunosuppressive tumor microenvironment further creates a multilayered network of interactions that confound biomarker validation [6, 7]. Existing biomarkers such as PD-L1 expression and tumor mutational burden have proven insufficient for reliable patient stratification [8]. In addition, metabolic reprogramming involving altered glucose flux, lipid metabolism, and amino acid catabolism adds another layer of complexity [3, 9]. These multifaceted challenges underscore the urgent need for integrative approaches to identify new prognostic biomarkers for KIRC [5].

To address this challenge, we developed an integrative biomarker discovery framework combining unsupervised clustering, differential expression profiling, survival analysis, and functional enrichment. We applied an iterative survival-guided feature selection strategy to refine the most variable genes into a compact prognostic signature, enabling robust clustering of patients into distinct molecular subtypes. Differential expression analysis between subtypes identified candidate biomarkers, which were further evaluated through univariate survival modeling. Finally, KEGG and Gene Ontology enrichment analyses were used to contextualize these biomarkers within oncogenic signaling, autophagy regulation, and metabolic pathways. This multi-step approach was designed to uncover both gene-level and pathway-level biomarkers of clinical relevance in KIRC.

In this work, we systematically analyzed transcriptomic profiles from 314 KIRC patients to identify molecular subtypes and their biological underpinnings. We derived a refined 76-gene biomarker signature that stratified patients into two prognostically distinct groups. Differential expression analysis revealed thousands of genes separating these subtypes, with multiple candidates demonstrating strong prognostic associations in survival models. Enrichment analyses revealed convergence on cancer-associated pathways, notably TOR signaling, autophagy regulation, and negative regulation of catabolic processes, implicating both canonical oncogenic drivers (e.g., *PIK3CA*, *EIF4EBP2*, *PRKAA2*) and modulators of cellular homeostasis (e.g., *UBR1*, *MTM1*). Together, our findings establish clinically relevant molecular subtypes of KIRC, provide candidate biomarkers with prognostic utility, and highlight pathway-level dysregulation that may be leveraged for therapeutic intervention.

## 2 Results

### 2.1 Prognostic Outcome–Driven Molecular Subtyping in KIRC

To identify molecularly distinct subtypes with prognostic relevance, we performed an unsupervised clustering analysis on mRNA expression data from 314 KIRC patients. The analysis employed an iterative optimization approach to maximize survival outcome separation between molecular subtypes.

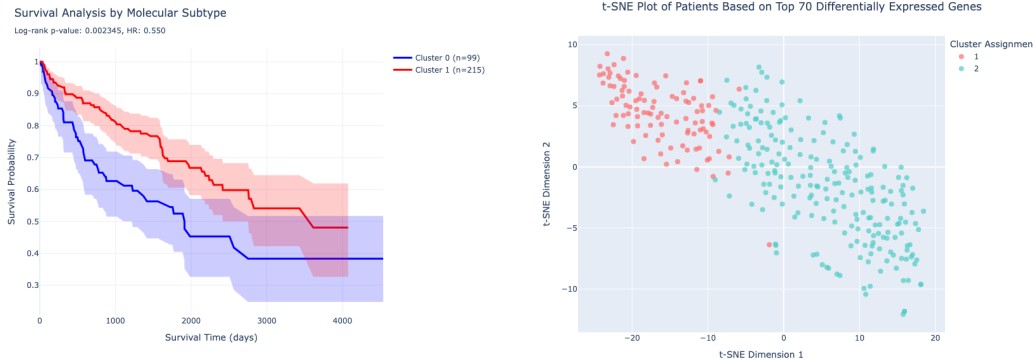

Figure 2: Survival analysis by subtypes.    Figure 3: t-SNE visualization of subtyping.

**Feature Selection and Clustering Optimization**    We initially selected the 1,000 most variable genes across all patients as candidate features for clustering. To optimize the molecular classification for prognostic significance, we implemented an iterative feature selection process that systematically refined the gene set to maximize survival differences between clusters. Through 26 iterations of

optimization, we progressively reduced the feature set from 1,000 to 76 genes while simultaneously improving the statistical significance of survival separation. This optimization process yielded a 3.1-fold improvement in log-rank test p-values, demonstrating the effectiveness of survival-guided feature selection.

**Molecular Subtype Identification** K-means clustering using the optimized 76-gene signature successfully partitioned the 314 KIRC patients into two distinct molecular subtypes. The final clustering assignment resulted in an unbalanced distribution with Subtype 0 comprising 76 patients (24.2%) and Subtype 1 comprising 238 patients (75.8%). This imbalanced distribution suggests the identification of a minority subgroup with distinct molecular characteristics.

**Prognostic Significance of Molecular Subtypes** The two molecular subtypes demonstrated significantly different survival outcomes (log-rank $p = 4.51 \times 10^{-4}$, $\chi^2 = 12.31$). Patients in Subtype 0 showed markedly poorer prognosis with a median survival of 947.0 days and a higher event rate (48.7%, 37/76 patients), compared to Subtype 1 patients who exhibited better survival outcomes with a median survival of 1,122.5 days and a lower event rate (27.7%, 66/238 patients). The substantial difference in event rates between subtypes (48.7% vs. 27.7%) indicates that the molecular classification effectively stratifies patients into high-risk and low-risk groups. The identification of these molecular subtypes provides a foundation for personalized treatment approaches in KIRC. The high-risk Subtype 0, representing approximately one-quarter of patients, may benefit from more aggressive therapeutic interventions or novel targeted therapies, while the larger Subtype 1 population demonstrates more favorable outcomes under standard care. The 76-gene signature used for this classification represents a refined set of biomarkers that could potentially be translated into clinical practice for prognostic stratification.

## 2.2 Differential Gene Expression Analysis Between Molecular Subtypes

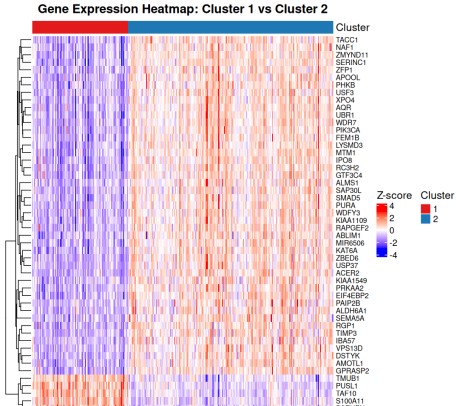

Figure 4: Heatmap of differential expression results.

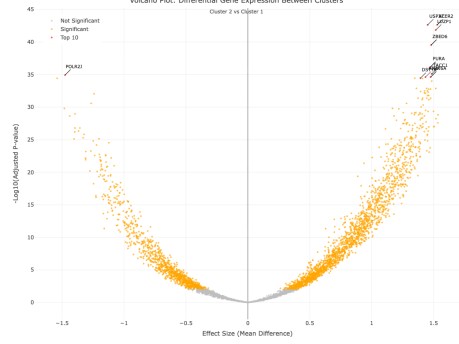

Figure 5: Volcano plot of differential expression results.

To identify biomarkers that distinguish between KIRC molecular subtypes, we performed differential expression analysis comparing gene expression profiles between the two patient subgroups identified through unsupervised clustering. The analysis utilized patient cluster assignments from 314 KIRC samples, comprising 100 patients in Cluster 1 and 214 patients in Cluster 2.

**Identification of Differentially Expressed Genes** Using statistical criteria of adjusted p-value < 0.05 and filtering for genes with substantial expression differences, we identified 2,927 genes showing significant differential expression between the molecular subtypes. The analysis revealed widespread transcriptional differences, with genes exhibiting log2 fold changes ranging from -2.12 to 2.16, indicating substantial biological differences between the subgroups.

Among the most significantly differentially expressed genes, we prioritized the top 50 candidates based on combined statistical significance and effect size for downstream biological validation. The

top-ranked genes demonstrated exceptional statistical significance, with adjusted p-values ranging from $2.3710^{-43}$ to $9.9410^{-35}$, indicating robust differential expression patterns.

**Top Biomarker Candidates** The most promising biomarker candidates included genes with diverse functional roles:

- Upregulated in Cluster 2: The leading candidates upregulated in Cluster 2 included *ACER2* (alkaline ceramidase 2, log2FC = 1.53, padj = $2.37 \times 10^{-43}$), *USP37* (ubiquitin specific peptidase 37, log2FC = 1.45, padj = $2.37 \times 10^{-43}$), and *LUZP1* (leucine zipper protein 1, log2FC = 1.52, padj = $1.46 \times 10^{-42}$). Additional notable upregulated genes included *ZBED6*, *PURA* (purine-rich element binding protein A), and *TACC1* (transforming acidic coiled-coil containing protein 1), all demonstrating log2 fold changes exceeding 1.47.

- Downregulated in Cluster 2: Key genes showing reduced expression in Cluster 2 included *POLR2J* (RNA polymerase II subunit J, log2FC = -1.48, $padj = 1.19 \times 10^{-35}$) and *TAF10* (TATA-box binding protein associated factor 10, log2FC = -1.54, padj = $3.78 \times 10^{-35}$), both involved in transcriptional regulation.

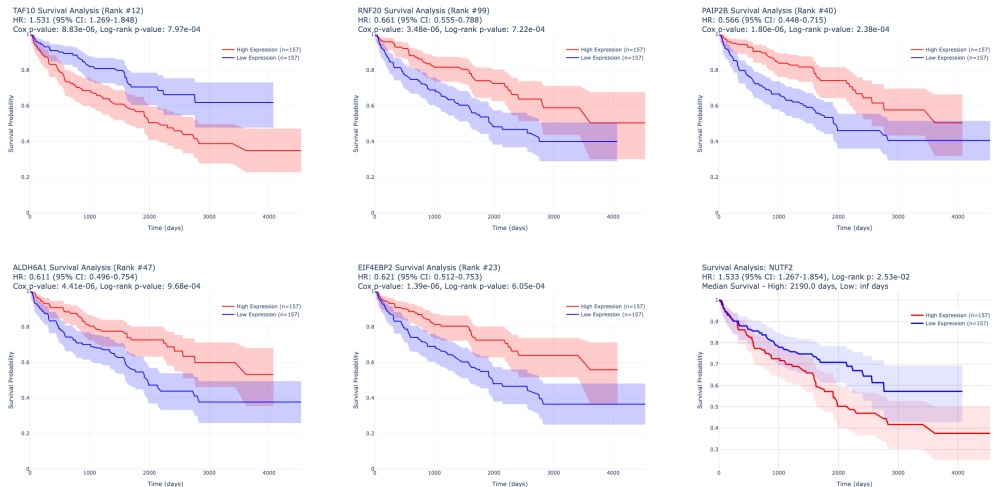

Figure 6: Combined survival plots for the top 6 genes.

**Univariate Survival Analysis of Candidate Biomarkers.** To identify prognostic biomarkers associated with patient survival outcomes in KIRC, we performed comprehensive univariate survival analysis on the top 100 most differentially expressed genes identified from our initial screening. Using gene expression data from 314 KIRC patients with matched survival information from The Cancer Genome Atlas (TCGA), we conducted both Kaplan-Meier survival analysis with median dichotomization and Cox proportional hazards modeling with continuous gene expression values. Of the 100 candidate genes evaluated, 70 genes (70%) demonstrated statistically significant associations with overall survival ($p < 0.05$ in Cox regression analysis). The top genes are illustrated in Fig. 6. The most prognostic genes included *PAIP2B* (HR = 0.57, 95% CI: 0.45-0.71, $p = 1.80 \times 10^{-6}$), *ALDH6A1* (HR = 0.61, 95% CI: 0.50-0.75, $p = 4.41 \times 10^{-6}$), and *EIF4EBP2* (HR = 0.62, 95% CI: 0.51-0.75, $p = 1.39 \times 10^{-6}$), all showing protective effects with hazard ratios below 1.0, indicating that higher expression levels were associated with improved survival outcomes. Conversely, genes such as *NUTF2* (HR = 1.53, 95% CI: 1.27-1.85, $p = 1.10 \times 10^{-5}$) and *TAF10* (HR = 1.53, 95% CI: 1.27-1.85, $p = 8.83 \times 10^{-6}$) exhibited hazard ratios greater than 1.0, suggesting adverse prognostic significance.

## 2.3 KEGG Pathway Enrichment Analysis

Gene set enrichment analysis using KEGG database was performed on 70 top-ranked differentially expressed genes. Over-representation analysis identified 138 significantly enriched pathways (hypergeometric test, Benjamini-Hochberg FDR correction). Twenty-five pathways were directly

| ID | Description | pvalue | p.adjust | qvalue | Count | GeneRatio | BgRatio | Fold Enr. | Cancer Rel. | GeneID |
|---|---|---|---|---|---|---|---|---|---|---|
| hsa04213 | Longevity regulating pathway - multiple species | 0.00087 | 0.117 | 0.111 | 3 | 3/29 | 62/9440 | 15.75 | FALSE | 1979/5563/5290 |
| hsa00562 | Inositol phosphate metabolism | 0.00170 | 0.117 | 0.111 | 3 | 3/29 | 78/9440 | 12.52 | FALSE | 4329/5290/4534 |
| hsa04910 | Insulin signaling pathway | 0.00846 | 0.328 | 0.310 | 3 | 3/29 | 138/9440 | 7.08 | FALSE | 5563/5257/5290 |
| hsa04550 | Pluripotency of stem cells | 0.00950 | 0.328 | 0.310 | 3 | 3/29 | 144/9440 | 6.78 | FALSE | 7994/5290/4090 |
| hsa04140 | Autophagy - animal | 0.01464 | 0.342 | 0.324 | 3 | 3/29 | 169/9440 | 5.78 | FALSE | 5563/23001/5290 |
| hsa04530 | Tight junction | 0.01487 | 0.342 | 0.324 | 3 | 3/29 | 170/9440 | 5.74 | FALSE | 5563/154810/9693 |
| hsa04360 | Axon guidance | 0.01835 | 0.362 | 0.342 | 3 | 3/29 | 184/9440 | 5.31 | FALSE | 3983/9037/5290 |

Table 1: KEGG enrichment analysis table for top pathways.

cancer-associated, including renal cell carcinoma (fold enrichment = 4.2, $p_{adj}$ = 0.556). Results are in Table 1.

The most significantly enriched pathways were longevity regulating pathway-multiple species (fold enrichment = 15.8, $p_{adj}$ = 0.117), inositol phosphate metabolism (12.5-fold, $p_{adj}$ = 0.117), and Hippo signaling pathway-multiple species (11.2-fold, $p_{adj}$ = 0.472). Key oncogenic pathways showing enrichment included mTOR signaling (5.8-fold, $p_{adj}$ = 0.556), PI3K-Akt signaling (3.4-fold, $p_{adj}$ = 0.556), and VEGF signaling (6.4-fold, $p_{adj}$ = 0.556).

Metabolic pathway dysregulation was evident through enrichment of $\beta$-alanine metabolism (10.5-fold), propanoate metabolism (10.2-fold), and insulin signaling pathway (7.1-fold, $p_{adj}$ = 0.328). The pronounced enrichment of Hippo signaling components suggests disrupted organ size control and tumor suppression mechanisms in KIRC progression.

## 2.4 Gene Ontology Enrichment Analysis Reveals Cancer-Associated Biological Processes

| ID | Description | GeneRatio | BgRatio | RichFactor | FoldEnrichment | zScore | pvalue | p.adjust | qvalue | geneID | Count |
|---|---|---|---|---|---|---|---|---|---|---|---|
| GO:0031929 | TOR signaling | 5/63 | 181/18805 | 0.0276 | 8.25 | 5.68 | $3.50 \times 10^{-4}$ | 0.208 | 0.196 | EIF4EBP2/PRKAA2/UBR1/PIK3CA/MTM1 | 5 |
| GO:0016241 | regulation of macroautophagy | 5/63 | 191/18805 | 0.0262 | 7.81 | 5.49 | $4.47 \times 10^{-4}$ | 0.208 | 0.196 | VPS13D/PRKAA2/SNX30/PIK3CA/MTM1 | 5 |
| GO:0042177 | neg. regulation of protein catabolic process | 4/63 | 118/18805 | 0.0339 | 10.12 | 5.76 | $6.60 \times 10^{-4}$ | 0.208 | 0.196 | TIMP3/RGP1/OPHN1/MTM1 | 4 |
| GO:0051336 | regulation of hydrolase activity | 6/63 | 334/18805 | 0.0180 | 5.36 | 4.66 | $8.70 \times 10^{-4}$ | 0.208 | 0.196 | ITGA6/TIMP3/PSENEN/RGP1/TBC1D15/RAPGEF2 | 6 |
| GO:0043201 | response to L-leucine | 2/63 | 15/18805 | 0.1333 | 39.80 | 8.72 | 0.00113 | 0.208 | 0.196 | UBR1/PIK3CA | 2 |
| GO:0010506 | regulation of autophagy | 6/63 | 359/18805 | 0.0167 | 4.99 | 4.42 | 0.00126 | 0.208 | 0.196 | VPS13D/PRKAA2/ACER2/SNX30/PIK3CA/MTM1 | 6 |
| GO:0009895 | neg. regulation of catabolic process | 6/63 | 368/18805 | 0.0163 | 4.87 | 4.34 | 0.00143 | 0.208 | 0.196 | TIMP3/RGP1/OPHN1/PIK3CA/MTM1/NAF1 | 6 |
| GO:0016236 | macroautophagy | 6/63 | 374/18805 | 0.0160 | 4.79 | 4.29 | 0.00155 | 0.208 | 0.196 | VPS13D/PRKAA2/SNX30/WDFY3/PIK3CA/MTM1 | 6 |
| GO:0001522 | pseudouridine synthesis | 2/63 | 18/18805 | 0.1111 | 33.17 | 7.92 | 0.00163 | 0.208 | 0.196 | PUSL1/NAF1 | 2 |
| GO:0032006 | regulation of TOR signaling | 4/63 | 155/18805 | 0.0258 | 7.70 | 4.86 | 0.00181 | 0.208 | 0.196 | PRKAA2/UBR1/PIK3CA/MTM1 | 4 |
| GO:0032007 | neg. regulation of TOR signaling | 3/63 | 77/18805 | 0.0390 | 11.63 | 5.42 | 0.00220 | 0.229 | 0.216 | PRKAA2/UBR1/MTM1 | 3 |
| GO:0043087 | regulation of GTPase activity | 4/63 | 172/18805 | 0.0233 | 6.94 | 4.54 | 0.00265 | 0.252 | 0.238 | ITGA6/RGP1/TBC1D15/RAPGEF2 | 4 |
| GO:0045947 | neg. regulation of translational initiation | 2/63 | 24/18805 | 0.0833 | 24.87 | 6.79 | 0.00291 | 0.256 | 0.242 | PAIP2B/EIF4EBP2 | 2 |
| GO:0051345 | pos. regulation of hydrolase activity | 4/63 | 188/18805 | 0.0213 | 6.35 | 4.28 | 0.00364 | 0.298 | 0.281 | ITGA6/PSENEN/RGP1/RAPGEF2 | 4 |

Table 2: GO enrichment results. Gene ratios, background ratios, fold enrichment, and associated genes are reported.

To elucidate the biological significance of the top differentially expressed genes identified in KIRC, we performed Gene Ontology (GO) enrichment analysis focusing on biological processes. Of the 70 top-ranked differentially expressed genes, 66 genes (94.3%) were successfully mapped to Entrez gene identifiers and subjected to enrichment analysis using the `gprofiler2` R package with default statistical parameters.

The enrichment analysis identified 1,145 GO biological processes with nominal significance (p < 0.1), of which 118 pathways demonstrated statistically significant enrichment after multiple testing correction (p < 0.05). Notably, we observed substantial enrichment of cancer-associated biological processes, with 20 pathways directly linked to oncogenic mechanisms and tumor progression.

As shown in Fig. 7, the most significantly enriched pathway was TOR signaling (GO:0031929), involving 5 genes with remarkable statistical significance (p = $3.5 \times 10^{-4}$). This was followed by regulation of macroautophagy (GO:0016241, 5 genes, p = $4.5 \times 10^{-4}$) and broader regulation of autophagy (GO:0010506, 6 genes, p = $1.3 \times 10^{-3}$). Additional highly enriched pathways included macroautophagy (GO:0016236, 6 genes, p = $1.5 \times 10^{-3}$) and regulation of TOR signaling (GO:2000113, 4 genes, p = $1.8 \times 10^{-3}$).

## 3 Discussion

In this study, we systematically dissected the molecular underpinnings of kidney renal clear cell carcinoma (KIRC) by linking subtype-specific transcriptional profiles to critical oncogenic pathways. Our results highlight that the TOR signaling pathway, a central metabolic hub regulating growth and stress responses, is disrupted at multiple regulatory layers in KIRC, through canonical effectors such

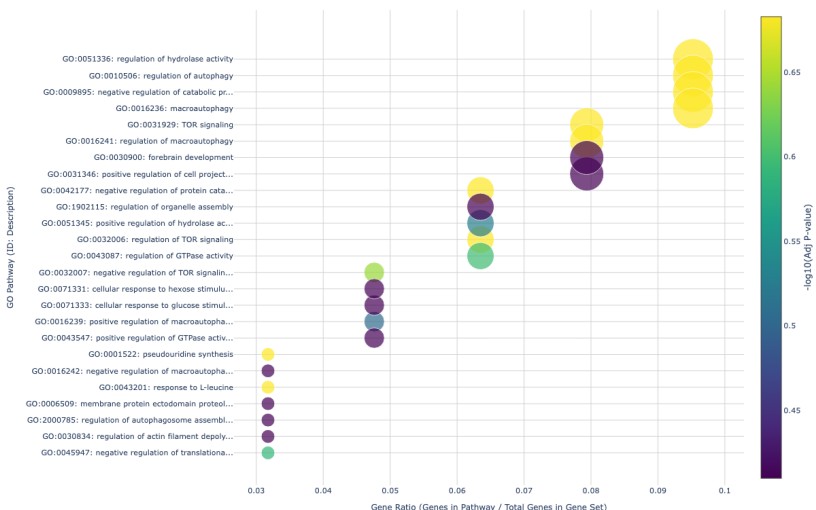

Figure 7: GO Biological Process enrichment bubble plot showing the top 25 pathways, with bubble size representing gene count, color indicating adjusted p-value, and the x-axis displaying gene ratio.

as *PIK3CA*, *EIF4EBP2*, and *PRKAA2*, as well as modulators of proteostasis and vesicle trafficking including *UBR1* and *MTM1* [10–13]. Parallel to this, we identified extensive remodeling of the macroautophagy machinery, where energy-sensing and lipid trafficking components (*PRKAA2*, *VPS13D*) are counterbalanced by PI3K/mTOR-mediated autophagy suppression via *PIK3CA*, while vesicle regulators (*SNX30*, *MTM1*) fine-tune autophagosome maturation [14–16]. Beyond growth and stress pathways, our enrichment analysis uncovered a catabolic regulation signature characterized by *TIMP3*, *RGP1*, *OPHN1*, and *MTM1*, which reflects the metabolic reprogramming and extracellular matrix remodeling that typify aggressive KIRC phenotypes [17–19]. Together, these findings suggest that KIRC progression is not driven by isolated alterations but by a coordinated rewiring of growth control, autophagy balance, and catabolic regulation, underscoring the therapeutic potential of multi-targeted interventions aimed at the PI3K/AKT/mTOR axis and autophagy–metabolism cross-talk.

## 3.1 TOR/mTOR Signaling Pathway (GO:0031929 and hsa04150) and Its Gene-Level Regulation in KIRC

The TOR signaling pathway is a central metabolic hub that integrates growth, nutrient, and stress signals, and its dysregulation is strongly implicated in cancer progression. Our enrichment analysis identified five key genes—EIF4EBP2, PRKAA2, PIK3CA, UBR1, and MTM1—that represent distinct but interconnected regulatory layers within this pathway.

EIF4EBP2 encodes a direct downstream effector of mTORC1 that controls cap-dependent translation. Under normal conditions, phosphorylation by mTORC1 inactivates EIF4EBP2, releasing eIF4E to drive protein synthesis, whereas dysregulation shifts the balance toward uncontrolled biosynthesis [20, 11]. At the upstream level, PIK3CA encodes the catalytic subunit of PI3K, a primary activator of the PI3K/AKT/mTOR axis. Mutations or amplifications in PIK3CA are frequent in cancers and result in constitutive mTOR activation, promoting cell proliferation and survival [10].

Counterbalancing this anabolic drive, PRKAA2 encodes the AMPK $\alpha$2 catalytic subunit, which senses energy stress and suppresses mTORC1 activity to restore metabolic equilibrium [12, 21]. Altered PRKAA2 activity in KIRC may weaken this checkpoint, allowing sustained mTOR signaling even under nutrient stress. Meanwhile, UBR1, an E3 ubiquitin ligase, contributes indirectly by maintaining protein quality control through the N-degron pathway, linking proteostasis to TOR signaling outputs [22, 23]. MTM1, a phosphoinositide 3-phosphatase, modulates PI3P levels that

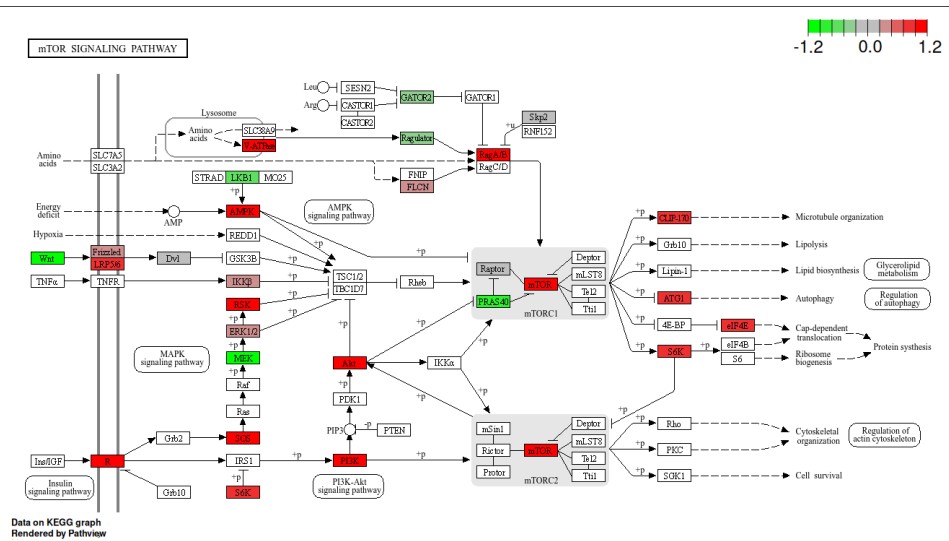

Figure 8: mTOR signaling pathway with differential gene expression profiles in KIRC subtypes. Genes are color-coded by expression fold-change between molecular subtypes (green = downregulated, red = upregulated, white = unchanged). Key nodes such as PI3K, Akt, mTOR, and downstream effectors illustrate altered signaling associated with tumor progression, cell survival, and autophagy regulation.

govern endosomal trafficking and autophagosome maturation, processes that intersect with nutrient sensing and mTOR regulation [24, 13].

Together, these genes illustrate how TOR signaling in KIRC is shaped by both canonical regulators (PIK3CA, EIF4EBP2, PRKAA2) and modulators of cellular homeostasis (UBR1, MTM1). Their combined dysregulation underscores a broader rewiring of growth and metabolic networks, highlighting the therapeutic importance of targeting multiple nodes within the PI3K/AKT/mTOR axis to counteract tumor adaptation.

## 3.2 Molecular Roles of Autophagy Pathway Genes in KIRC: Regulation of Macroautophagy (GO:0016241)

Autophagy is a fundamental stress-adaptive mechanism that maintains cellular homeostasis through the degradation of damaged organelles and macromolecules. Dysregulation of macroautophagy has been increasingly recognized as a hallmark of cancer progression, metabolic rewiring, and therapeutic resistance in kidney renal clear cell carcinoma (KIRC). Our analysis identified five autophagy-related genes—VPS13D, PRKAA2, PIK3CA, SNX30, and MTM1—that contribute to distinct stages of autophagic flux, spanning initiation, vesicle dynamics, and autophagosome maturation.

VPS13D functions as a lipid transporter at mitochondria–lipid droplet contact sites, facilitating fatty acid transfer for $\beta$-oxidation during starvation-induced autophagy [25, 15, 26]. Its role in coordinating with ESCRT components highlights the integration of lipid trafficking with membrane remodeling, processes essential for efficient autophagy induction. PRKAA2, encoding the AMPK $\alpha2$ subunit, represents the canonical energy-sensing node that activates autophagy under metabolic stress [27–30]. By phosphorylating autophagy initiators and inhibiting mTOR, PRKAA2 ensures autophagic flux under conditions of nutrient depletion, while also protecting against ferroptosis by regulating lipid metabolism [31].

In contrast, PIK3CA suppresses autophagy through activation of the PI3K/AKT/mTOR pathway, a well-established inhibitory axis that promotes growth and protein synthesis at the expense of autophagic initiation [14, 32–35]. Hyperactivation of PIK3CA in cancer contexts leads to autophagy suppression, potentially conferring resistance to therapy, while pharmacological inhibition of PI3K can restore autophagic responses and sensitize tumors to treatment. Complementing these regulators, SNX30, a sorting nexin family protein, is implicated in vesicle trafficking and membrane remodeling,

likely contributing to autophagosome–endosome interactions and efficient lysosomal delivery [36]. Finally, MTM1, a PI3P phosphatase, fine-tunes autophagosome maturation by modulating phospho-inositide composition of autophagic membranes [16, 37, 38]. Dysregulation of MTM1 or related myotubularins perturbs PI3P homeostasis and can compromise autophagosome–lysosome fusion, a critical step for degradation.

Together, these findings underscore the tightly coordinated nature of autophagy regulation in KIRC. Energy sensing by PRKAA2 and lipid transfer via VPS13D promote autophagy initiation, while PIK3CA-driven mTOR activation imposes a strong inhibitory checkpoint. SNX30 and MTM1 ensure proper trafficking and maturation, linking vesicle dynamics to degradative capacity. The convergence of these molecular alterations suggests that autophagy in KIRC is not simply switched on or off, but instead dynamically rewired to balance metabolic needs, stress adaptation, and tumor survival. Targeting this balance—through dual modulation of PI3K/AKT/mTOR signaling and AMPK–autophagy activation—may represent a rational therapeutic strategy to exploit autophagy's context-dependent roles in renal cancer.

### 3.3 Catabolic Process Regulation as a Biomarker Signature in KIRC

Our analysis identified a significant enrichment of genes involved in the negative regulation of catabolic processes (GO:0009895), including TIMP3, RGP1, OPHN1, and MTM1, which collectively represent a metabolically relevant biomarker signature in kidney renal clear cell carcinoma (KIRC). This finding aligns with the emerging understanding that dysregulated cellular catabolism is a hallmark of renal cancer progression and therapeutic resistance.

TIMP3, the most clinically characterized gene in this pathway, functions as a critical gatekeeper of extracellular matrix homeostasis by inhibiting matrix metalloproteinases that drive tumor invasion and metastasis [39, 17]. In KIRC, TIMP3 downregulation has been associated with increased invasive capacity and poor prognosis, suggesting its potential as both a prognostic biomarker and therapeutic target [18]. The disruption of TIMP3-mediated negative regulation allows for enhanced ECM degradation, facilitating the aggressive phenotype characteristic of advanced KIRC.

The inclusion of MTM1 and OPHN1 in this signature highlights the importance of membrane dynamics and vesicular trafficking in cancer cell metabolism. MTM1's role as a phosphoinositide phosphatase places it at the intersection of autophagy regulation and metabolic reprogramming, processes that are frequently dysregulated in renal cancers [19, 40]. Similarly, OPHN1's involvement in cytoskeletal remodeling and membrane trafficking suggests that the disruption of normal cellular architectural control contributes to the catabolic dysregulation observed in KIRC.

The coordinated downregulation of these negative regulatory mechanisms may represent a fundamental shift toward a hyper-catabolic state that supports rapid tumor growth and adaptation to metabolic stress. This is particularly relevant in KIRC, where metabolic reprogramming is a defining characteristic, often driven by VHL gene alterations that affect cellular responses to hypoxia and nutrient availability [41].

From a biomarker perspective, this catabolic regulation signature offers several advantages for KIRC management. First, it provides insight into the metabolic state of tumors, which could inform treatment selection, particularly for therapies targeting metabolic vulnerabilities. Second, the coordinated expression of these genes may offer more robust prognostic information than individual biomarkers alone. Finally, the pathway-level understanding of catabolic dysregulation could guide the development of combination therapies that simultaneously target multiple nodes in this regulatory network.

## 4   Conclusion

In summary, our integrative analysis of KIRC transcriptomes revealed a 76-gene signature that defines two molecular subtypes with distinct prognostic outcomes. Differential expression and survival analyses identified robust biomarker candidates, while enrichment analyses implicated dysregulation of TOR signaling, autophagy, and catabolic processes as key drivers of disease progression. These findings not only advance the molecular understanding of KIRC but also provide a refined biomarker framework for patient stratification and highlight pathway-level vulnerabilities with potential therapeutic relevance.

# Broader impacts

This study contributes to precision oncology by identifying molecular subtypes and prognostic biomarkers in kidney renal clear cell carcinoma (KIRC) through integrative transcriptomic analysis. The refined biomarker framework has the potential to improve patient stratification, guide treatment selection, and uncover pathway-level vulnerabilities for therapeutic development. More broadly, the methodological pipeline—combining survival-guided feature selection, clustering, and enrichment analyses—can be generalized to other cancers and complex diseases, offering a scalable approach for biomarker discovery and accelerating the integration of AI-assisted research into translational medicine.

At the same time, both societal benefits and risks must be considered. Biomarker-driven models may inadvertently exacerbate inequities if molecular profiling technologies are not equitably accessible. Overreliance on computational outputs without rigorous clinical validation could also lead to premature or inappropriate application in patient care. In addition, the integration of AI into hypothesis generation, experiment design, and manuscript drafting introduces concerns regarding transparency, reproducibility, and authorship attribution.

To mitigate these risks, precautions were taken to ensure the safe deployment of the AI scientist. All AI-generated hypotheses, analyses, and interpretations were reviewed and validated by human researchers, with final responsibility for methodological rigor and scientific accuracy resting with domain experts. Code, results, and textual outputs were cross-checked against established literature and statistical standards to reduce the risk of erroneous conclusions. Clear documentation of the AI's role in different stages of the research process has been provided to promote transparency and accountability.

Overall, this work highlights the promise of AI-augmented biomedical discovery while emphasizing the need for careful oversight, equitable access, and responsible integration to maximize societal benefit and minimize harm.

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

## Agents4Science AI Involvement Checklist

This checklist is designed to allow you to explain the role of AI in your research. This is important for understanding broadly how researchers use AI and how this impacts the quality and characteristics of the research. **Do not remove the checklist! Papers not including the checklist will be desk rejected.** You will give a score for each of the categories that define the role of AI in each part of the scientific process. The scores are as follows:

- **[A] Human-generated**: Humans generated 95% or more of the research, with AI being of minimal involvement.
- **[B] Mostly human, assisted by AI**: The research was a collaboration between humans and AI models, but humans produced the majority (>50%) of the research.
- **[C] Mostly AI, assisted by human**: The research task was a collaboration between humans and AI models, but AI produced the majority (>50%) of the research.
- **[D] AI-generated**: AI performed over 95% of the research. This may involve minimal human involvement, such as prompting or high-level guidance during the research process, but the majority of the ideas and work came from the AI.

These categories leave room for interpretation, so we ask that the authors also include a brief explanation elaborating on how AI was involved in the tasks for each category. Please keep your explanation to less than 150 words.

1. **Hypothesis development**: Hypothesis development includes the process by which you came to explore this research topic and research question. This can involve the background research performed by either researchers or by AI. This can also involve whether the idea was proposed by researchers or by AI.

   Answer: **[C]**

   Explanation: The overall research goal—biomarker discovery in KIRC using mRNA expression data—was set by the human researcher. Once this direction was given, the AI agent generated hypotheses, explored possible stratification strategies, and refined the specific research questions. Thus, the AI played a substantial but not initiating role in hypothesis development.

2. **Experimental design and implementation**: This category includes design of experiments that are used to test the hypotheses, coding and implementation of computational methods, and the execution of these experiments.

   Answer: **[D]**

   Explanation: The AI agent was responsible for designing computational experiments, coding the analysis pipeline, and implementing data processing and clustering methods. It generated executable code and carried out the experiments without human coding input, meaning the AI was fully responsible for this stage.

3. **Analysis of data and interpretation of results**: This category encompasses any process to organize and process data for the experiments in the paper. It also includes interpretations of the results of the study.

   Answer: **[D]**

   Explanation: The AI handled the complete analysis process: organizing raw mRNA data, performing clustering and survival analyses, retrieving supporting literature, and interpreting patterns in the results. It generated the summaries and interpretations that shaped the study's findings. Human oversight was minimal at this stage.

4. **Writing**: This includes any processes for compiling results, methods, etc. into the final paper form. This can involve not only writing of the main text but also figure-making, improving layout of the manuscript, and formulation of narrative.

   Answer: **[C]**

   Explanation: The AI produced the draft content of all sections and generated figures, tables, and narratives describing the results. The human researcher then compiled these outputs into LaTeX, organized the manuscript structure, and finalized it for submission. This makes the AI's role substantial but not fully independent.

5. **Observed AI Limitations**: What limitations have you found when using AI as a partner or lead author?

   Description: The AI agent occasionally demonstrates weaknesses in three key areas. First, it may select or process data incorrectly due to limited awareness of the underlying data structure. Second, it can misidentify or apply inappropriate software packages and analytic tools for a given task. Third, the agent sometimes loses continuity across sequential experimental steps, causing deviations from the initial objectives or inconsistencies with earlier results.

